# Comparison of polypeptides that bind the transferrin receptor for targeting gold nanocarriers

Conor McQuaid[¤a], Andrea Halsey[¤b], Maëva Dubois, Ignacio Romero, David Male*

Department of Life Health and Chemical Sciences, The Open University, Milton Keynes, United Kingdom

¤a Current address: Department of Neuroscience, University of Rochester, Rochester, New York, United States of America
¤b Current address: Department of Neuroscience and Ophthalmology, Institution of Inflammation and Ageing, College of Medical and Dental Sciences, University of Birmingham, Edgbaston, Birmingham, United Kingdom

* David.Male@Open.ac.uk

**Data Availability Statement:** All relevant data are within the paper and its Supporting information files.

**Funding:** DM, PhD studentship to support CM. 'Delivery of therapeutic cytokines to the CNS, using

## Abstract

The ability to target therapeutic agents to specific tissues is an important element in the development of new disease treatments. The transferrin receptor (TfR) is one potential target for drug delivery, as it expressed on many dividing cells and on brain endothelium, the key cellular component of the blood-brain barrier. The aim of this study was to compare a set of new and previously-described polypeptides for their ability to bind to brain endothelium, and investigate their potential for targeting therapeutic agents to the CNS. Six polypeptides were ranked for their rate of endocytosis by the human brain endothelial cell line hCMEC/D3 and the murine line bEnd.3. One linear polypeptide and two cyclic polypeptides showed high rates of uptake. These peptides were investigated to determine whether serum components, including transferrin itself affected uptake by the endothelium. One of the cyclic peptides was strongly inhibited by transferrin and the other cyclic peptide weakly inhibited. As proof of principle the linear peptide was attached to 2nm glucose coated gold-nanoparticles, and the rate of uptake of the nanoparticles measured in a hydrogel model of the blood-brain barrier. Attachment of the TfR-targeting polypeptide significantly increased the rates of endocytosis by brain endothelium and increased movement of nanoparticles across the cells.

## Introduction

The targeting of therapeutic agents to specific cells or tissues is central to the development of treatments for many diseases. Increasing the proportion of a drug that reaches the target-tissue reduces potential off-target effects and the dose required to produce a therapeutic effect. This latter point is particularly important as expensive biological agents are increasingly being developed for clinical use. The main approach to targeting has been to identify receptors that

nanoparticle carriers.' Midatech pharma plc. https://www.midatechpharma.com The funders had no role in study design, data collection and analysis, decision to publish, or preparation of the manuscript.

**Competing interests:** Conor McQuaid was supported by a PhD studentship from Midatech Pharma plc, awarded to DM (2015-2018). This does not alter our adherence to PLOS ONE policies on sharing data and materials. The company has no patents, products in development or marketed products related to the material presented in this paper. None of the authors is employed either directly or as a consultant by Midatech Pharma plc. None of the authors has ever applied for employment with the company.'

are expressed on the target cell and develop antibodies, peptides or aptamers that bind to that receptor and which may be coupled to a therapeutic agent [1–4].

The transferrin receptor (TfR) is a particularly attractive target as it is expressed on many dividing cell-types and has therefore often been chosen for targeting tumour cells [5, 6]. In addition, the TfR is highly expressed on brain capillary endothelium, where it acts as an essential transporter of iron to the brain [7–9]. Expression on other vascular endothelial cells is relatively low [10]. Delivering therapeutics to the brain is especially problematic because of the blood-brain barrier (BBB) [3, 11]. The majority of drugs with potential to treat brain diseases cannot cross the barrier [12] and none of the large therapeutic biomolecules (cytokines, antibodies, siRNA etc), can pass through the tight junctions between brain endothelial cells [13]. As the TfR enables endocytosis and transcytosis of transferrin (Tf) a large serum protein (80kDa), it could also theoretically be used to transport therapeutic agents into the brain [14]. It has been debated as to what proportion of the TfR undergoes transcytosis, is recycled to the apical membrane or is degraded in endosomal compartments. Indeed, the intracellular fate of the Tfr partly depends on the bound ligand. Nevertheless this receptor does have potential for transport of large therapeutic biomolecules.

Several targeting systems have been developed that aim to use the TfR for transport to the brain, including antibodies, antibody fragments and peptides [7, 14, 15]. These targeting agents have been isolated from libraries by a variety of techniques, including direct-binding to the TfR. Interestingly divalent antibodies with high affinity for the TfR are not necessarily most effective in promoting transcytosis, as the receptor can be diverted to lysosomal pathways by the binding of high-affinity ligands and/or cross-linking [16]. Ideally a reagent that targets the TfR should be small, easy to attach to the cargo-molecules or nanoparticles that are being transported and not interfere with endosomal transport. Polypeptides can fulfil many of these requirements [17] and various approaches have been made to develop brain-targeting peptides [18–20].

The aim of this study was to compare the properties of a number of different peptides that bind to the TfR, including a linear peptide [21] and three recently-described cyclic peptides isolated from a phage-display library by binding to murine and human TfR [22]. It was particularly notable that one of these cyclic peptides has primary sequence homology with a cyclic peptide isolated by binding to brain endothelium *in vivo*, from a much older study [23].

This work compares the rate of endocytosis of the polypeptides after binding to the human brain endothelial cell line hCMEC/D3. One important consideration in developing peptides that bind the TfR is that they should still be active in the presence of serum molecules, including Tf itself. Ideally Tf and the polypeptide would bind separate sites on the TfR. This would also ensure that using the peptide to aid drug transport would reduce potential interference with the normal iron transport system of the brain. Hence another element of the study was to assess peptide interaction with brain endothelium in the presence of different amounts of serum or Tf. The study has identified important properties and distinct differences between the peptides tested, each of which could be advantageous for targeting the TfR for endocytosis or transcytosis.

## Materials and methods

### Peptides

The polypeptides were produced by Peptide Synthetics (Fareham, UK) and purity confirmed by mass spectrometry (>95%). Cyclic peptides were tagged with Fitc at the N-terminus and the linear peptide at the C-terminus, to allow tracking. Spacers (GA) were added to the original

**Table 1. Characteristics of peptides.**

| Peptide | Sequence | Structure | Source |
|---------|----------|-----------|--------|
| Pep-1 | Fitc-GAWSIIDCSMNYCLYIEG | Cyclic, disulphide bond C-C | [22] |
| Pep-2 | Fitc-GAIHCHPQGDQSVSFCWR | Cyclic, disulphide bond C-C | |
| Pep-10 | Fitc-GALHECTYYWWGLDCSFR | Cyclic, disulphide bond C-C | |
| Pep-10-M | Fitc-GALHEDTYYWWGLDKSFR | Cyclic, peptide bond between D-K | New |
| Pep-R1 | Fitc-GACENWWGDVCGAGAG | Cyclic, disulphide bond C-C | [23] |
| Pep-R2 | Fitc-GACLSSRLDACGAGAG | Cyclic, disulphide bond C-C | |
| Pep-L | THRPPMWSPVWPSK-Fitc | Linear,Fitc attached to K side chain | [21] |
| Pep-L* | THRPPMWSPVWPCS | -CS added for exchange reaction | New |

sequences in order to reduce the possibility that the Fitc would interfere with TfR-binding (Table 1).

Additional data on these peptides is given in S1 Table. Peptides were dissolved in dimethyl sulfoxide at 10mg/ml and stored in aliquots at -80˚C until use at 1–10µg/ml in Hank's balanced salt solution (HBSS) with the stated amounts of foetal bovine serum (FBS).

## Cell cultures

The human brain endothelial line hCMEC/D3, originally characterised in this laboratory [24] was used at passage 24–33 and cultured in modified EBM-2 MV medium (Lonza) containing (v/v) 0.025% VEGF, IGF and EGF, 0.1% bFGF, 0.1% rhFGF, 0.1% gentamycin, 0.1% ascorbic acid and 0.04% hydrocortisone, according to the manufacturer's instructions and 1% 100 U/ml penicillin and 100µg/ml streptomycin. The cells were split at 90% confluency onto fresh collagen-coated flasks or plates (12 or 24 well) for experiments, with medium changed every 2-3days.

Human microvascular endothelial cells (MVEC) originally derived from lung were obtained from ScienCell and grown in the same conditions as stated above. bEnd.3 mouse brain endothelial cells were supplied by the European culture collection and grown in DMEM with 10%FBS, penicillin and streptomycin. All cell cultures were maintained at 37˚C in air with 5% $CO_2$.

## Cytometry and analysis of peptide endocytosis

hCMEC/D3, bEnd.3 or MVEC (Lonza, CC-2527) cells were grown to confluence in a 12-well or 24-well plate and washed 2x with HBSS at 37˚C before being treated with stated concentrations of each peptide (equal fluorescence intensity). Treated cells were incubated at 37˚C for 30mins–3hrs. They were then washed twice with warmed HBSS and detached from their wells by adding 500µl of trypsin + EDTA solution (Sigma-Aldrich, Dorset, UK, Cat No. T3924) and incubating at 37˚C for 5mins. The contents of each well were harvested into individual 15ml Falcon tubes. The wells were washed for remaining cells by adding another 1ml of HBSS, which was then also added to the respective tubes. The tubes were centrifuged for 5mins at 4˚C and 1500 RPM. All but the cell pellet was then removed and the cells washed in 1ml cold HBSS.

The cells were centrifuged under the same conditions and all but the cell pellet removed. The cells were resuspended in 300µl HBSS and the total solution transferred to chilled FACS tubes. The cytometer (FACscan, Becton & Dickinson) was set to capture 10,000 cells in the FSC/SSC gate, and the gain voltage on the FL1 detector (fluorescein) adjusted so that 90% of

untreated cells gave a FL1 value <10. The median fluorescence values were taken from each culture as a single data point.

Each sample was analysed in triplicate (separate cultures) and each experiment carried out 2–4 times with concordant results. FACS histograms are representative and the data compiled to show the mean ±SEM of the median fluorescence of each of the triplicate cultures.

## Nanoparticles and quantitation of gold

Gold nanoparticles (NP), obtained from Midatech Pharma Plc, were synthesised using a modified Brust-Schiffrin method and capped during synthesis with thiol-C2-glucose [25]. Gold in the NPs was measured by the following spectrophotometric method in 96 well plates. A total volume of 10μl of sample was applied to each well. To this, 30μl of 50% fresh, cold Aqua Regia (mixture of conc. nitric acid and conc. hydrochloric acid in a 1:3 Molar ratio) was applied. The liquid was mixed by gentle tapping and left to stand for 1 minute to atomise the gold. Next, 150μl of 2M NaBr was added. The absorbance was read on a plate reader OPTIMA FluoSTAR at 382nm against a gold standard (Sigma). Each sample and the standards were tested in triplicate.

## Ligand exchange reaction

A ligand exchange reaction occurs when AuNPs (with a coating of thiol-linked sugar residues) are mixed with ligands containing a free thiol group, in aqueous solution. In this case, 10.2nmol of NP (defined as 100 gold atoms/core) was combined with the appropriate ratio of ligand, ie in the 1:1.5 exchange reaction mixture, 15.3nmol of peptide, Pep-L (1722MW) = 2.6μl of 10mg/ml stock or Pep-L+Fitc (2080MW) = 3.2μl of 10mg/ml stock, in a 15ml falcon tube. This mixture was vortexed 15-30secs and then incubated at 37˚C at 600rpm for 3hrs in a PHMT Grant-bio Thermo Fisher heat/shaker block. All samples were then spun at 17,000G for 30secs after which samples were spun-filtered 3 times in 15ml 10kDa MWCO vivaspins at 4000G and resuspended in ddH$_2$O.

## Transport assay

Transport of the nanoparticles was measured using a 3-dimensional hydrogel model of the blood-brain barrier in 24-well plates (1cm$^2$, Greiner). The hydrogels were prepared with 2.0mg/ml type 1 rat tail collagen (First Link, Wolverhampton, UK) in 0.6% acetic acid in water. The gels (0.5ml per well) were set by neutralisation with 1M NaOH and equilibrated with MEM. When set, hCMEC/D3 cells were seeded at a density of 80,000–100,000 cells per well and incubated for 2–3 days in culture medium. After the monolayer was formed, the endothelial cells were switched to a medium without VEGF for 3 days to form tight junctions. The nanoparticles were applied to the surface of the endothelium at a final concentration of 8μg/ml (gold) and incubated for 3 hours at 37˚ C. After the incubation, the media was removed, cells washed 2x with HBSS and the endothelial cells removed with a brief collagenase digestion (1mg/ml) until the monolayer detached from the collagen (~10mins). The collagen gel was then fully digested with collagenase to recover nanoparticles that had crossed the endothelium. The endothelium and collagen digest were analysed for gold by ICP-Q-MS. This culture system has the advantage that nanoparticles released from the basal surface of the endothelium (after transcytosis) are not trapped on a filter or in the basal lamina, but diffuse away from the endothelium.

## Statistical analysis

All FACS analyses were carried out with independent cell cultures in triplicate and median fluorescence value of 10,000 cells from each culture was measured. Data is shown as the mean

of the median fluorescence, ±SEM. Statistical analysis was carried out using Prism-7 software, with ANOVAR followed by Dunnett's multiple comparison test, or Student's t-test as appropriate. Each experiment was repeated 3 times with concordant results.

## Results

A set of peptides known to bind to the TfR and peptides that selectively bind brain endothelium were synthesised with a fluorescent tag (Fitc) on either the N- or C-terminal amino acids (Table 1). Specific fluorescence of the peptides was measured by fluorimetry; to facilitate comparison between the peptides they were applied at concentrations producing equal fluorescence in the tracking studies (S1 Table). In addition an equivalent amount of Fitc-dextran (70kDa) was included as a control for non-specific endocytosis (ie not receptor mediated). These peptides were applied to the human brain endothelial cell line hCMEC/D3 to establish an appropriate time-point and concentrations for later experiments. Peptides were first incubated with the endothelium for 0–3 hours. Unbound peptides were removed, the endothelial cells released by trypsinisation and the endocytosed peptides detected and quantitated by flow cytometry. All FACS histograms showed a single peak, and median fluorescence values were derived from each histogram. Data points represent the mean (n = 3) of the median fluorescence values. The time course of internalisation is shown in Fig 1A.

The results in all cases showed a progressive rate of uptake, which was fastest in the first hour, but the values from each peptide differed considerably. The time course did not saturate (no plateau), so the 3hr time-point was chosen for subsequent experiments, as it gave the largest differential between different peptides. A comparison of peptide uptake at 3 hours showed that Pep-1, Pep-10 and Pep-L showed highest rates of endocytosis (Fig 1B). Pep-2 also showed significant uptake, but the two peptides isolated by *in vivo* binding (Pep-R1 and Pep-R2), were not consistently significantly higher than the Fitc-dextran control. Note that the FACS data measures the amount of accumulated endocytosed peptide, not the position of the Tfr, so the results do not give information on the subcellular location of the receptor or its traffic pathways through the cell.

Some variation in the absolute values was seen in independent experiments, but the overall ranking of the peptide-uptake was always Pep-L, Pep-1, Pep-10 > Pep-2> Pep-R1,Pep-R2.

Peptide endocytosis by hCMEC/D3 cells was then compared with uptake by another human microvascular endothelial cell line (MVEC) and a mouse brain endothelial cell line, bEnd.3 (Fig 2). For transport studies, it is important that targeting peptides can bind to both human and mouse TfR. This allows *in vitro* studies on human cells to be followed by *in vivo* studies in rodents. The results show that Pep-1, Pep-2 and Pep-10 also bind to mouse brain endothelium and the MVECs. (It has previously been reported that Pep-L also recognises mouse TfR [21]. However, there were some notable differences in the binding of individual peptides to different cell types: Pep-1 and Pep-R2 both bound relatively strongly to MVECs and Pep-10 bound more strongly to the mouse line than to the two human cell lines. Two factors may explain these results. MVEC, which are rapidly dividing cells, express high levels of the TfR as detected by cell-surface ELISA and Pep-10 has been shown to bind significantly more strongly to mouse TfR than human TfR [22].

The binding of Pep-1, Pep-2, Pep-10, and Pep-L to hCMEC/D3 was examined in more detail. The initial studies (Figs 1–3) were carried out with peptides at equivalent fluorescence to facilitate comparison of the peptides by FACS (S1 Table). Uptake of peptides was directly proportional to the concentration applied in the range 1–30µg/ml (S1 Fig) indicating that the mechanism of internalisation does not saturate at these concentrations.

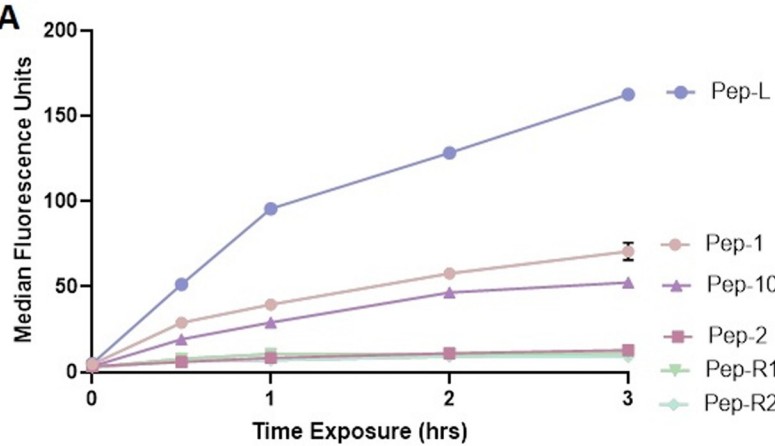

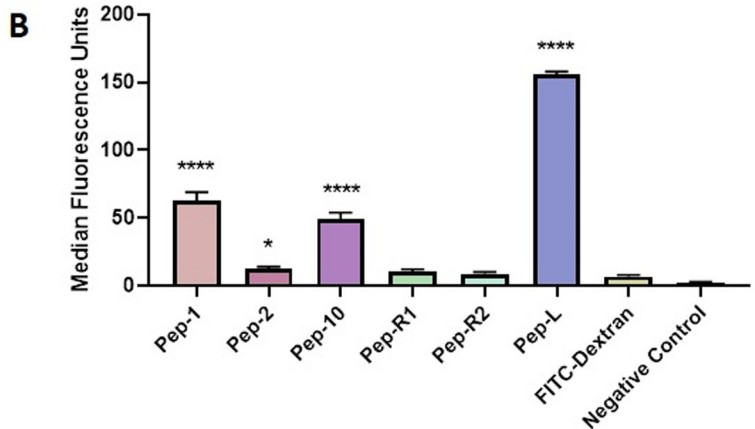

**Fig 1. Endocytosis of peptides by hCMEC/D3 cells.** A. Uptake of peptides by hCMEC/D3 cells over 3 hours measured by FACS. Values are mean ± SEM of the median fluorescence from 3 replicate cultures. B. Mean ± SEM of the median fluorescence of 6 peptides binding to hCMEC/D3 cells. Data is derived from 3 biological replicates and 3 technical replicates of each culture, analysed by Dunnet's multiple comparison test, comparing each peptide with the FITC-dextran non-specific endocytosis control. * P<0.05, **** P<0.0001.

The uptake assays were then repeated in the presence of variable levels of bovine serum albumin (BSA) or human Tf (Fig 3). Uptake of Pep-L was strongly inhibited in the presence of 8mg/ml serum albumin, a concentration approximately equivalent to 20% of the level in serum. Pep-L has a high proportion of hydrophobic amino acids and a net positive charge (S1 Table), so it is likely that binding to negatively charged serum albumin reduces the effective concentration of free-peptide available for binding to the TfR.

Pep-10 showed dose-dependent inhibition of binding to the endothelium in the presence of Tf; 1mg/ml Tf is a concentration similar to that found in normal serum. The result suggests that Pep-10 binds to the TfR close to the Tf binding site, but there was still some uptake, even at the highest level of Tf. Pep-1 uptake was weakly inhibited with Tf. Pep-L was also partly inhibited by Tf, but did not show a simple dose-dependent effect.

In a separate set of experiments (S2 Fig) it was confirmed that treatment of hCMEC/D3 cells with Tf did not alter expression of the TfR on the membrane surface of hCMEC/D3 cells.

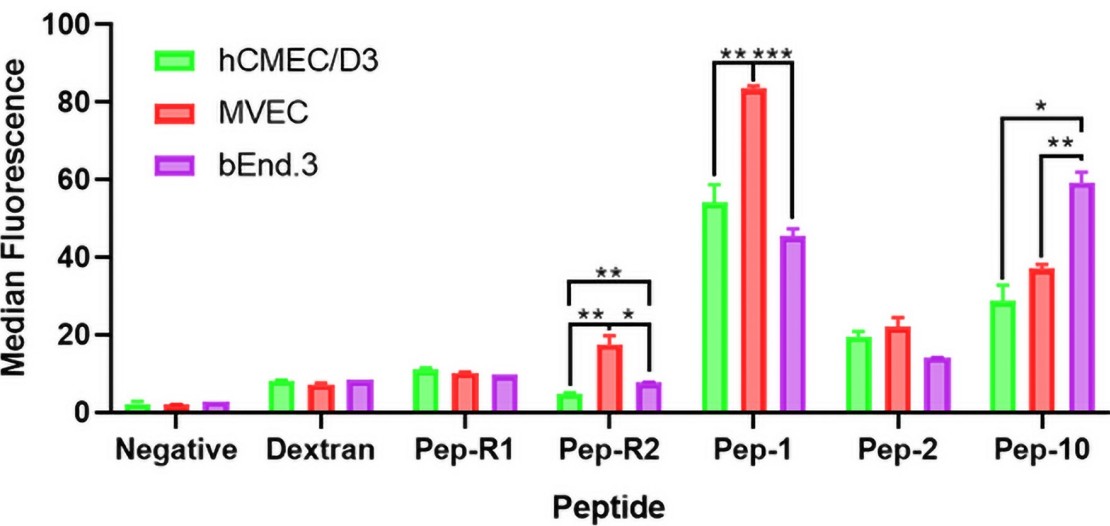

**Fig 2. Endocytosis of TfR peptides by different endothelial cell lines.** Endocytosis of peptides by hCMEC/D3, MVEC and bEnd.3 cells analysed by FACS. Data shown is the mean ±SEM of the median fluorescence of 3 biological replicates. Binding of each peptide on different cell types was compared using an unpaired t-test. (*** = $p<0.001$, ** = $p<0.01$ and * = $p<0.05$).

Hence the inhibition of Pep-10 uptake caused by Tf was not due to a loss of the TfR, but competition for the receptor.

One potential use of the TfR-targeting peptides is to improve uptake of nanocarriers into cells or tissues expressing the TfR. Binding of cargo molecules or peptides onto gold nanoparticles is readily effected by the exchange reaction, in which a ligand with a free–SH group exchanges, in reducing conditions, with a ligand bound to the gold core by Au-S bonds. However cyclic peptides with a loop held by a disulphide bond between cysteine residues cannot be used in the exchange reaction, because the conditions required would break open the loop and the free–SH groups could then attach to the nanoparticle, thereby destroying the peptide structure. In order to circumvent this limitation, one possible solution is to substitute the Cys-Cys bridge holding the loop with an alternative chemical bond, that is stable in reducing conditions. To determine if this approach was feasible a variant of Pep-10 was synthesised (Pep-10M) in which the two Cys residues were substituted by aspartic acid (D) and lysine (K) residues, cross-linked by a peptide bond between their side chains (Table 1). The uptake of Pep-10M by hCMEC/D3 cells was then compared with the original Pep-10 (Fig 4). The FACS histogram demonstrated that the modified peptide could still bind to the cells, although it was less effective than the original peptide.

From the initial experiments Pep-L appeared to have the best potential for targeting to brain endothelium. To demonstrate the potential, a variant of Pep-L was synthesised with the addition of a cysteine residue near the C-terminus, to allow it to engage in an exchange reaction (Table 1, Pep-L*). The modified peptide had the two C-terminal residues, replaced with -CS to allow attachment to the NP and the Fitc (if present) placed on the N-terminus. The peptide was attached via the exchange reaction to 2nm core diameter gold nanoparticles with a thiol-C2-glucose ligand shell. In the reaction, the cysteine of the Pep-L* displaced some of the C2-glucose residues. The reaction was performed with two different molar ratios of nanoparticle:peptide (in these conditions, 1–4 ligands may exchange with the C2-glucose on each nanoparticle) and free peptide was removed on a spin filter. The nanoparticles were applied to a model of the blood-brain barrier with hCMEC/D3 cells on a collagen gel [26]. This model allows free diffusion of nanoparticles away from the basal membrane of the endothelium and

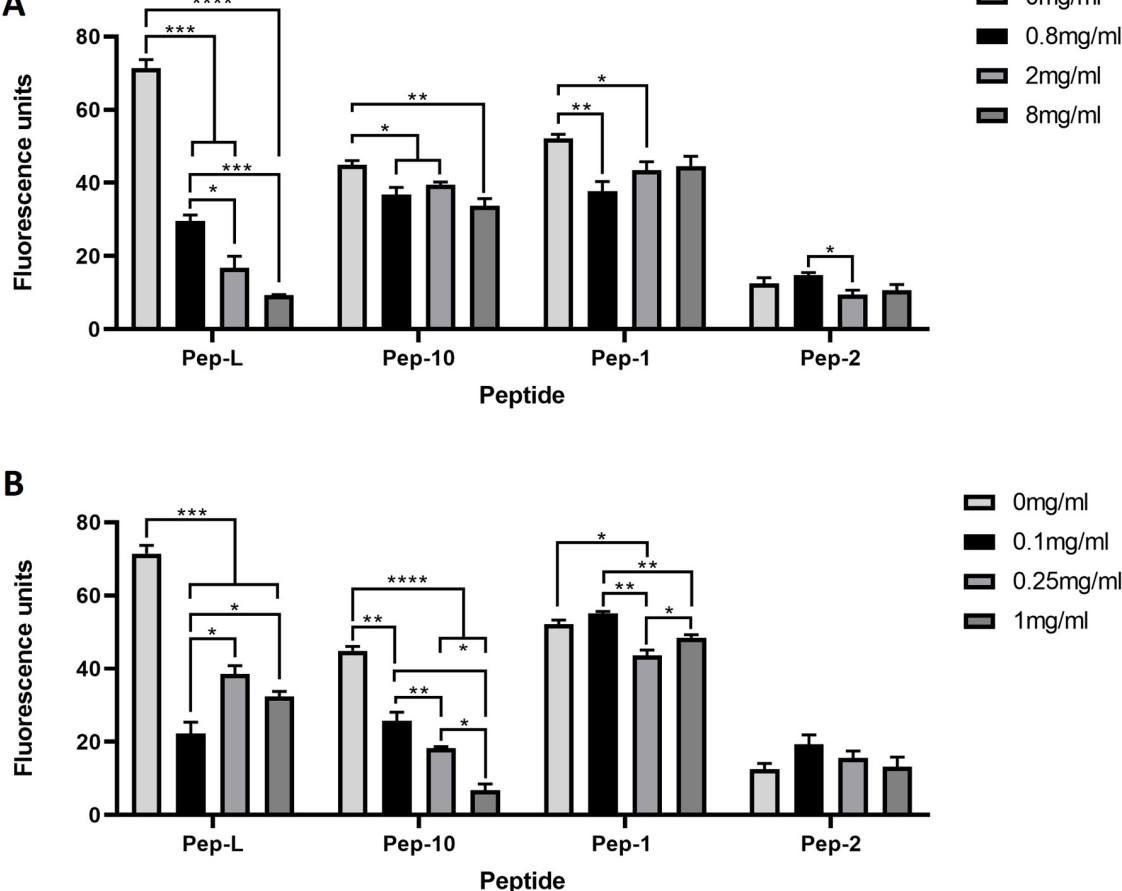

**Fig 3. Inhibition of peptide uptake by BSA and transferrin.** Endocytosis of TfR-binding peptides by hCMEC/D3 cells in the presence of different amounts of BSA (A) or transferrin (B). Data are the mean ±SEM (n = 3) of the median fluorescence of independent cultures. Data was analysed by Dunnet's multiple comparison test. * $p < 0.05$, ** $p < 0.01$, ***$p < 0.001$, **** $p < 0.0001$.

electron micrographs demonstrate that these gold glyconanoparticles cross the endothelium by a combination of vesicular and cytosolic transcytosis, with the relative amounts moving by each route, dependent on the formulation of the nanoparticle and cargo molecules [27]. The amount of gold recovered in the cells and in the gel is shown in Fig 5. At the ratio of NP:peptide 1:1.5, significantly more gold was found in both the cell layer and the collagen gel. Although this system was not optimised, it demonstrates that the addition of a TfR targeting peptide can enhance endocytosis and transcytosis of a gold nanocarrier across the brain endothelial cells.

## Discussion

The ability to target a therapeutic agent to specific cells or tissues is a major goal in the development of effective treatments for disease. Polypeptides are particularly attractive for this purpose, as they can be synthesised in quantity and at high purity. They are also relatively small in comparison with antibody and antibody-fragments used for targeting receptors or in comparison with the nanocarriers currently under development. Cyclic polypeptides with 4–8 amino-acid residues in a loop closed by a disulphide bond between two cys residues have been developed against a number of receptors including the TfR [21, 22]. Typically these polypeptides are

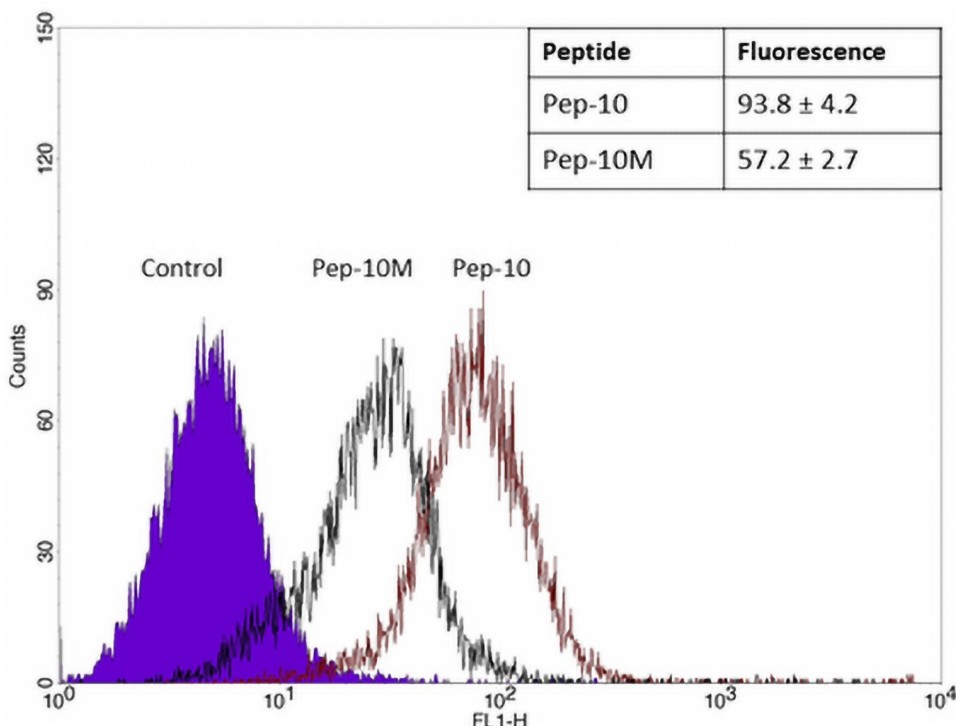

| Peptide | Fluorescence |
|---------|--------------|
| Pep-10 | 93.8 ± 4.2 |
| Pep-10M | 57.2 ± 2.7 |

**Fig 4. Structural modification of TfR-binding Pep-10.** A representative FACS histogram comparing uptake of Pep-10 and Pep-10M by hCMEC/D3 cells after 3 hours. The inset table shows the mean ± SEM of the median fluorescence of 3 replicate cultures.

10–20 amino acid residues in length (1 – 3kDa) and cyclisation gives structural stability to the polypeptide loop that binds to the receptor on the target cell. Cyclisation also increases their resistance to degradation by peptidases.

The TfR, which is selectively expressed on brain endothelium (TfR-2) has been identified for its potential to transport therapeutic agents across the blood-brain barrier [14, 28]. In this paper we have investigated four polypeptides originally isolated for their binding to human and mouse TfR (Pep-1, Pep-2, Pep-10, Pep-L). Two polypeptides that selectively bound to brain endothelium *in vivo* were also investigated (Pep-R1, Pep-R2). Since Pep-R2 has homology (-WWG-) in the loop with Pep-10, it is likely that Pep-R2 also binds to the TfR. Several other peptides isolated by binding to TfR also have this motif in the centre of the loop [22].

Comparison of the binding of the different peptides to the human brain endothelial cell line hCMEC/D3 demonstrated strong uptake of Pep-1, Pep-10 and Pep-L and lower uptake of Pep-2 (Fig 2). Pep-R1 and Pep-R2 showed weak uptake which was not consistently significantly higher than the dextran control for (non-specific) absorptive endocytosis. Further examination of the four TfR-binding peptides, showed that albumin inhibited endocytosis of Pep-L (Fig 3A) which is the most hydrophobic of the set and the only peptide with a positive charge at neutral pH. Binding to negatively-charged albumin by electrostatic interaction would reduce the effective concentration of available free Pep-L, and could therefore explain reduced endocytosis in the presence of albumin. While this may be an advantage in extending the half-life of a peptide-based therapeutic agent [29] it could prove to be a limitation for the use of Pep-L as a targeting molecule in serum and other tissue fluids that have high amounts of albumin. However, when attached to a nanoparticle additional considerations apply. The surface ligands of nanoparticles considerably affects their uptake by endothelial cells in two major

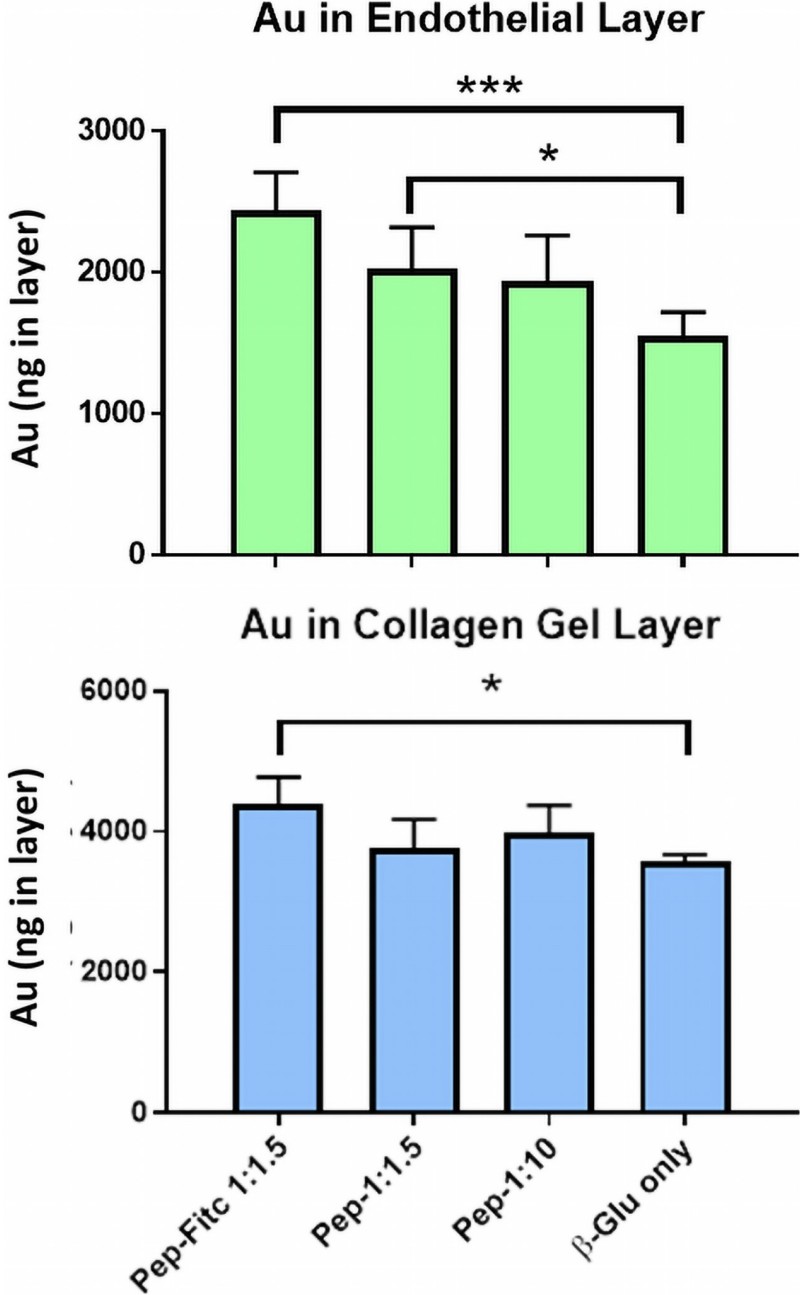

**Fig 5. Transcytosis of gold NPs with targeting peptide.** Recovery of gold in the hCMEC/D3 cells or the collagen gel, after application of gold NPs with different concentrations of Pep-L* (with or without Fitc) to the apical surface of hCMEC/D3 cells. Values are mean ± SEM of 6 biological replicates, each with two determinations of gold concentration. Nanoparticles with bound peptides were compared with C2-glucose-NPs (β-Glu) by paired t-test (* $P < 0.05$, *** $P < 0.001$).

ways: acidic or basic groups attract a corona of serum proteins of the opposite charge which may effectively flip the charge of the nanoparticle [30] and bound proteins can enhance uptake by the endothelium for both receptor mediated transport (RMT) and absorptive mediated transport (AMT). Additionally an increase in charge of nanoparticles (positive or negative) generally increases uptake by the endothelium [31, 32].

Of the remaining peptides, Pep-10 endocytosis was inhibited and Pep-1 slightly inhibited by Tf at concentrations normally found in serum (Fig 3B). Neither peptide was completely blocked at 1 mg/ml Tf which is approximately a 4-fold molar excess of Tf:peptide. The affinity of Tf with bound iron for the TfR is high and binding induces global changes in the conformation of the ectodomain of the receptor [33]. Consequently Tf binding could alter affinity of the peptides for TfR even if they bind at sites distant from the Tf-binding site. This is the most likely explanation for the reduction in binding of Pep-1 and Pep-10 in the presence of free Tf.

Peptides (1-3kDa) have some distinct advantages over larger targeting molecules such as IgG antibodies (~160kDa) or antibody fragments. Their small size is less likely to alter the characteristics of a nanocarrier, and they cannot cross-link the receptor. Cross-linking of receptors often results in trafficking of the receptor through endosomal compartments to lysosomes and degradation. For delivery of therapeutic agents it is important that the targeted receptor remains in the recycling/transcytosis pathways. For this reason monovalent antibody fragments targeting the Tfr appear to be more effective than whole antibody in delivering material to the brain [34].

While there are several advantages in the use of targeting peptides, there are potential disadvantages; firstly the binding site is small by comparison with, for example, an antibody and consequently the affinity of the peptide for a receptor is likely to be lower than the affinity of an antibody; For transcytosis across brain endothelium it has been shown that high affinity antibodies are less effective than medium affinity antibodies [35, 36], and reducing antibody to the Tfr can enhance its level of transcytosis [16]. Hence the lower affinity of peptides may actually be an advantage in targeting therapeutic agents to the brain.

Another possible limitation of these peptides is that the disulphide bond that cyclises them can be cleaved in reducing conditions. To test whether this could be overcome, a variant of Pep-10 was produced, in which the disulphide bond between Cys residues was replaced with a peptide bond between Asp and Lys residues. The length of the cross-link is longer than the disulphide bond, so it was anticipated that this would affect the configuration of the loop and consequently binding affinity to the TfR and endocytosis by the endothelium. The structural modification did indeed reduce binding, but only partially (Fig 4) which indicates that more stable variants of the cyclic peptides could be synthesised and still retain TfR-binding activity. Other chemistries for cyclising peptides have been developed and could be substituted where the disulphide bond presents problems, as in the exchange reaction described here. However, any alteration in the size of the loop or the length of the cross-bridge is likely to affect binding of the peptide to the receptor.

As proof of principle Pep-L was attached to 2nm gold nanoparticles (NP) by exchange with the C2-glucose coat on the NPs. At the higher molar ratio in the exchange reaction, the peptide significantly enhanced NP-endocytosis by the brain endothelium and increased the amount that had crossed the endothelium and localised in the collagen matrix beneath the basal membrane (Fig 5). Clearly, any use of peptides for targeting therapy would require optimisation for each therapeutic agent. For example, gold NPs can carry more than one peptide moiety, which could increase their avidity for the TfR. Also, the affinity of binding of the agent to the TfR could affect its intracellular trafficking. In the case of brain endothelium the aim is to promote transcytosis, and moderate affinity binding is likely to be advantageous, since the peptide must release from the TfR if it is to carry material across the endothelium. Conversely, if one used the peptides to target a cytotoxic agent to the TfR on a tumour cell, high affinity binding would be preferred in order to maximise internalisation of the agent in the target cell.

## Conclusions

Four peptides, Pep-1, Pep-2, Pep-10 and Pep-L all have potential for targeting the human and mouse transferrin receptor (TfR). The linear peptide, Pep-L is inhibited by albumin, which may limit its utility in tissue fluids with high levels of albumin. The cyclic peptides, Pep-1, Pep-2 and Pep-10 bind to both human and mouse TfR and are taken up at different levels of efficiency by endothelial cells expressing the TfR. Pep-1 and Pep-10, which show highest levels of endocytosis, appear to bind near the Tf-binding site on the receptor. Modification of Pep-10 by replacing the disulphide bond of the loop with a peptide bond, reduces the susceptibility to reducing conditions, enabling ligand exchange and increasing stability *in vivo*. A peptide with a single free thiol can be attached to gold nanoparticles by a simple exchange reaction and this enhances transcytosis across brain endothelium and hence may be used to increase delivery of therapeutic agents across the blood-brain barrier.

## Supporting information

**S1 Table. Specific fluorescence and concentrations of peptides in culture, net charge at pH 7 and proportion of hydrophobic residues.**
(DOCX)

**S1 Fig. Dose dependency of endocytosis of Pep-10 and Pep-L by hCMEC/D3 cells.**
(DOCX)

**S2 Fig. Expression of TfR on hCMEC/D3 cells measured by cell-surface ELISA.**
(DOCX)

## Author Contributions

**Conceptualization:** David Male.

**Data curation:** Conor McQuaid, David Male.

**Funding acquisition:** David Male.

**Investigation:** Conor McQuaid, Andrea Halsey, Maëva Dubois, David Male.

**Methodology:** Conor McQuaid, David Male.

**Project administration:** David Male.

**Supervision:** Ignacio Romero, David Male.

**Writing – original draft:** Conor McQuaid, David Male.

**Writing – review & editing:** Conor McQuaid, David Male.

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
