## [Decision Letter · Decision Letter 0]

30 Mar 2021

PONE-D-21-03180

Comparison of polypeptides that bind the transferrin receptor for targeting gold nanocarriers.

PLOS ONE

Dear Dr. Male,

Thank you for submitting your manuscript to PLOS ONE. After careful consideration, we feel that it has merit but does not fully meet PLOS ONE’s publication criteria as it currently stands. Therefore, we invite you to submit a revised version of the manuscript that addresses all the points raised during the review process.

Three experts have evaluated the paper and agreed that it is of interest for the field, but needs improvements as detailed in the reviews. Among others the methods need to be completed (e.g. statistical analysis) and the discussion need to be modififed based on the suggestions.

We look forward to receiving your revised manuscript.

Kind regards,

Mária A. Deli, M.D., Ph.D.

Academic Editor

PLOS ONE

Journal Requirements:

"DM, PhD studentship to support CM.

'Delivery of therapeutic cytokines to the CNS, using nanoparticle carriers.'

Midatech pharma plc.

https://www.midatechpharma.com

The funders had no role in study design, data collection and analysis, decision to

publish, or preparation of the manuscript."

We note that you received funding from a commercial source: "Midatech pharma plc."

Reviewers' comments:

Reviewer's Responses to Questions

**Comments to the Author**

1. Is the manuscript technically sound, and do the data support the conclusions?

Reviewer #1: Yes

Reviewer #2: Yes

Reviewer #3: Yes

2. Has the statistical analysis been performed appropriately and rigorously? 

Reviewer #1: Yes

Reviewer #2: Yes

Reviewer #3: Yes

3. Have the authors made all data underlying the findings in their manuscript fully available?

Reviewer #1: Yes

Reviewer #2: Yes

Reviewer #3: Yes

4. Is the manuscript presented in an intelligible fashion and written in standard English?

Reviewer #1: Yes

Reviewer #2: Yes

Reviewer #3: Yes

5. Review Comments to the Author

Reviewer #1: In the current manuscript “Comparison of polypeptides that bind the transferrin receptor for targeting gold nanocarriers”, the authors investigated six peptides, Pep-1, Pep-2, Pep-R1, Pep-R2, Pep-10 and Pep-L all have potential for targeting the human and mouse transferrin receptor (TfR). This work compared the rate of endocytosis of these polypeptides after binding to human brain endothelial cells, hCMEC/D3, human microvascular endothelial cells (MVEC) and mouse brain endothelial cells (bEnd.3). The study demonstrated that four peptides, Pep-1, Pep-2, Pep-10 and Pep-L bound to both human and mouse TfRs and were taken up by the cells at different levels. The linear peptide, Pep-L was inhibited by bovine serum albumin, which limits its utility in serum condition. The authors supposed that Pep-1 and Pep-10, with highest levels of endocytosis, appeared to bind near the Tf-binding site on the receptor. Finally, the authors modified Pep-L, which could be attached to gold nanoparticles by a simple exchange reaction and this improved the transcytosis of nanocarriers across the BBB.

My findings are the following:

1. Regarding the ”Materials and methods” part, the exact marking of the catalog number of the MVEC (Line 103) is missing, please correct it.

2. The ”Statistical Analysis” point is absent from the methods, authors should complete it.

3. Line 245: ”The nanoparticles were applied to a model of the blood-brain barrier with hCMEC/D3 cells on a collagen gel (27). ” I lack of the description of this BBB model from the ” Materials and methods” part. It would be more understandable if authors add this method to the manuscript.

4. Line 187: ”Pep-1 and Pep-R1 both bound relatively strongly to MVECs…”

In Fig. 3, the Pep-1 and Pep-R2 seems to bind strongly to MVECs, please clarify this.

5. It is not clear, why Pep-L is not present in Fig. 3, please explain it?

6. Finally, both the Pep-10 and Pep-L were chemically modified for gold nanoparticles, but only the Pep-L coupled with nanocarriers was tested on the BBB model, please clarify it.

In conclusion, the experiments are well designed and the whole study is consistent and easy to follow. The conclusions are moderate. After minor revision, I recommend the acceptance of the manuscript for publication.

Reviewer #2: The manuscript by McQuaid et al., analyze the endocytosis of several peptides targeting the transferrin receptor. Most of them have been publish before in peer-reviewed papers or in a local thesis from the authors lab. Two of the peptides have been made in a new version in order to optimize for coupling to gold-particles. The aspect of find and analyze peptide with capability to be transported into the brain (over the BBB) is highly relevant and needed by the society to improve drug delivery. However, the amount of new data is limited and would benefit from being increased before publication. Particular kinetic data and transcytosis data are needed.

My comments and suggestions are:

The aspect of TfR as a transcytosing receptor is important for this work. It has been questioning by many if Tf actually is transcytosed across the BBB and the data demonstrating transport of Tf is limiting. This is shortly mentioned in line 59-60 and statement is concluded on ref 14. Ref 14 is minor non peer received book chapter and not sufficient as reference for this discussion. In the light of the importance for the observed result this should be more intensely introduced/discussed. Please elaborate on this and find better evidence.

Line 39-40 (abstract); The provided data do not demonstrate any evidence of transcytosis as stated. This statement should be removed or demonstrated by a tight model (see also comment below).

Figure two is one time point taken from figure 1, just show as a histogram. Not necessary to display in two figures, at most it should be an A and B panel in one figure. Moreover, Fig 3 does not provide much to the paper and is the data form Fig 1 and 2 tested on other cells

Why is PEP10-M and Pep-L* not included in figure 1,2,3 and 4? Fif 5 show that it does influence on uptake, and it would be relevant for the data in figure 6 and the discussion.

A major lack of the article is the absence of kinetic data. More fluorescence in the cells measured by FACS could be a result of fast/low recycling, detaching from receptor, degradation. Statement of more strong binding (e.g. line 187 and other places) cannot be used here. Only binding data obtained from solid phase binding studies or isothermal titration calorimetry and provide these information’s. Also, the peptide has been ranked according to the rate of endocytosis (mentioned several times in the abstract). Again, I believe that more data regarding recycling possible transcytosis and degradation are needed to make these conclusions. Please convince me or rephrase.

What timepoint is used in figure 3 and later figures? And Why is Pep-L not included here?

Is it acceptable to use submitted material and Thesis as references?

Which are the two cyclic peptide that is referred to in the abstract (Pep-L, Pep-L*, Pep10 or Pep10-M). Pep-L* and Pep10-M should be included in figure 4

Line 215-216: “Pep-L was also partly inhibited by Tf, but…” It seems to me that that higher amount of Tf increase the fluorescence? The “No-addition” of BS and Tf of all peptides should be included in the figure

Line 234: “..it was slightly less effective..” With respect to what? Endocytosis I believe should be written. Moreover “slightly” is not the right word to use here, it seems to be 50 %.

Please introduce the collagen gel model in the materials and methods. What is beta-Glu? It is not introduced anywhere and why is it used as a control? The author should somehow provide evidence that their model is tight by use of a gold particle labeled with an irrelevant peptide or dextran of similar size. EM images could be relevant here as a supplement.

Figure 5 and 6 should also include a comparison of Pep-L and Pep-L* and Pep10 and pep10*. It does not make sense to compare the endocytosis of Pep10/pep10* in fig 5 and then use the L variant for “transcytosis” studies

Line 301-302; Transcytosis of TfR antibodies is perhaps not only dependent on affinity. The avidity also seems to be an important issue as demonstrated by several papers from Ryan Watts group and Per-Ola Fresgaard group. Avidity might be a factor an important for nanoparticles and it will be a nice aspect to include in the discussion.

Not all the peptides could be inhibited by Tf. Have the authors any idea whether these peptides can bind to other receptors?

Pep10 has been published in a local thesis. Are there any evidence that this peptide in fact binds to TfR or is it just an assumption? If there are could some of these data be included in this paper?

Reviewer #3: The authors investigated six polypeptides for their endocytosis and transcytosis in human brain endothelial line CMEC/D3 and mouse brain endothelial line B.end3 and the interference of albumin and transferrin on the endocytosis. The authors tagged gold nanoparticles with one of the peptides (the linear peptide) and observed increased uptake by the endothelial cells. The authors concluded that the attachment of TfR-targeting polypeptide significantly increased the rate of endocytosis and transcytosis by the endothelial cells. The experiments were well conducted and the data were well analyzed and presented. The following comments may need to be considered by the authors.

1. For the uptake assays via endocytosis, the authors did not describe how the background noise was subtracted. A control peptide (such as a scrambled peptide) should be included in the experiments to determine the random uptake of the peptide by the endothelial cells in comparison with specific TfR-mediated uptake.

2. Binding of the polypeptides to TfR: the authors did not provide any direct evidence that their polypeptides can physically interact with and bind to TfR although transferrin was shown to inhibit endocytosis of some polypeptides tested. The authors may need to consider performing SPR analysis or cross-link of the polypeptides to TfR or Co-IP with TfR to determine whether their polypeptides indeed physically bind to TfR.

3. Transcytosis assay: It is well known that the in vitro BBB models constituted by CMEC/D3 or B.end3 cell lines are very leaky. The authors did not describe how the collagen gel model was performed and whether there was any control molecule or control peptide to control the permeability and leakage of the model used.

4. A minor point is that the authors said in the manuscript that TfR is specifically expressed in brain endothelial cells. This is not true. A recent paper published in FBCNS analyzed the expression patterns of TfR and other RMT receptors in human and mouse brain vessels and peripheral tissues/vessels. The study found that TfR is also expressed in peripheral tissues and vessels. Thus, it is more accurate to say that TfR is more enriched in brain endothelial cells or expressed at a higher level in brain endothelial cells.

6. PLOS authors have the option to publish the peer review history of their article (what does this mean?). If published, this will include your full peer review and any attached files.

Reviewer #1: No

Reviewer #2: No

Reviewer #3: No

---

## [Author Response · Author response to Decision Letter 0]

29 Apr 2021

Reviewer #1: 

1. Regarding the ”Materials and methods” part, the exact marking of the catalog number of the MVEC (Line 103) is missing, please correct it. The cell source has been added.

2. The ”Statistical Analysis” point is absent from the methods, authors should complete it. A paragraph on Statistical analysis has been added to the M&M section

3. Line 245: ”The nanoparticles were applied to a model of the blood-brain barrier with hCMEC/D3 cells on a collagen gel (27). ” I lack of the description of this BBB model from the ” Materials and methods” part. It would be more understandable if authors add this method to the manuscript. A section on the transport system has been added to the M&M section. 

4. Line 187: ”Pep-1 and Pep-R1 both bound relatively strongly to MVECs…”

In Fig. 3, the Pep-1 and Pep-R2 seems to bind strongly to MVECs, please clarify this. This mistake has been corrected. (We thank the reviewer for their close attention to the text.)

5. It is not clear, why Pep-L is not present in Fig. 3, please explain it? The experiments in figure 3 were done before those described in figures 1 and 2. Pep-L was added to the study following the information in Figure 3. Since the binding of Pep-L to human and mouse endothelium had already been published [Ref 21], repeating the experiments with Pep-L would add no further information.

6. Finally, both the Pep-10 and Pep-L were chemically modified for gold nanoparticles, but only the Pep-L coupled with nanocarriers was tested on the BBB model, please clarify it. A sentence has been added to clarify why the specific modification (removal of -K-Fitc and addition of -CS) was made.

Reviewer #2: 

My comments and suggestions are:

The aspect of TfR as a transcytosing receptor is important for this work. It has been questioning by many if Tf actually is transcytosed across the BBB and the data demonstrating transport of Tf is limiting. This is shortly mentioned in line 59-60 and statement is concluded on ref 14. Ref 14 is minor non peer received book chapter and not sufficient as reference for this discussion. In the light of the importance for the observed result this should be more intensely introduced/discussed. Please elaborate on this and find better evidence. Reference 14 has been replaced by a primary reference that discusses transcytosis of the Tfr and its potential for use as a transport carrier. Two additional sentences has been added to the text.

Line 39-40 (abstract); The provided data do not demonstrate any evidence of transcytosis as stated. This statement should be removed or demonstrated by a tight model (see also comment below). The word ‘transcytosis’ has been removed from the abstract and replaced with a more precise term.

Figure two is one time point taken from figure 1, just show as a histogram. Not necessary to display in two figures, at most it should be an A and B panel in one figure. Moreover, Fig 3 does not provide much to the paper and is the data form Fig 1 and 2 tested on other cells. Figures 1 and 2 have been combined into a two-part figure, A and B. The data in figure 3 is very important. It is essential to show that the peptides bind to both the human and mouse Tfr so that studies can be done in vivo in rodents in parallel with in vitro studies with human cells, using the same peptides. This point is made in the text.

Why is PEP10-M and Pep-L* not included in figure 1,2,3 and 4? Fif 5 show that it does influence on uptake, and it would be relevant for the data in figure 6 and the discussion. Pep10M was produced after the initial set of studies (identifying suitable peptides) since the unmodified peptide was not suitable for use in the exchange reaction. Pep-L* could not be used in the FACS studies (Figs 1-4) because it lacks a fluorescent tag.

A major lack of the article is the absence of kinetic data. More fluorescence in the cells measured by FACS could be a result of fast/low recycling, detaching from receptor, degradation. Statement of more strong binding (e.g. line 187 and other places) cannot be used here. Only binding data obtained from solid phase binding studies or isothermal titration calorimetry and provide these information’s. Also, the peptide has been ranked according to the rate of endocytosis (mentioned several times in the abstract). Again, I believe that more data regarding recycling possible transcytosis and degradation are needed to make these conclusions. Please convince me or rephrase. There is some kinetic data in figure 1A. However, in order to do the FACS, the endothelium is detached by trypsinisation to form a single cell suspension and the FACS measures endocytosed fluorescent peptide. It does not show the position of the Tfr, (As single cell suspensions, the cells no longer have an apical and basal membrane and many cell surface molecules are removed by trypsinisation.) This point has been added to the text. 

What timepoint is used in figure 3 and later figures? Three hours as stated in the text (line 183). 

Is it acceptable to use submitted material and Thesis as references? The thesis is published open access and obtainable from http://oro.open.ac.uk/55108/ This has been added to the reference list. It was hoped that the submitted paper would be published by the time this article was reviewed, but it is still undergoing revision so it has been removed, together with associated text.

Which are the two cyclic peptide that is referred to in the abstract (Pep-L, Pep-L*, Pep10 or Pep10-M). Pep-L* and Pep10-M should be included in figure 4 See reply to point above about the modified peptides and why they were not included in the FACS studies.

Line 215-216: “Pep-L was also partly inhibited by Tf, but…” It seems to me that that higher amount of Tf increase the fluorescence? The “No-addition” of BS and Tf of all peptides should be included in the figure. Figure 4A and 4B have been replaced with new figures that include the zero-dose point. (This now figure 3A and 3B). The addition of the extra dose-point does not change the fundamental result or conclusions of this experiment.

Line 234: “..it was slightly less effective..” With respect to what? Endocytosis I believe should be written. Moreover “slightly” is not the right word to use here, it seems to be 50 %. ‘slightly’ has been deleted.

Please introduce the collagen gel model in the materials and methods. What is beta-Glu? It is not introduced anywhere and why is it used as a control? The author should somehow provide evidence that their model is tight by use of a gold particle labeled with an irrelevant peptide or dextran of similar size. EM images could be relevant here as a supplement. A section has been added on the model. We have previously shown that nanoparticles move across the endothelium by vesicular transcytosis and via cytosolic diffusion. Those papers include many EM images - one additional reference has been added to this paper and a short addition to the text.. 

Figure 5 and 6 should also include a comparison of Pep-L and Pep-L* and Pep10 and pep10*. It does not make sense to compare the endocytosis of Pep10/pep10* in fig 5 and then use the L variant for “transcytosis” studies Pep-L was chosen because previous studies had shown that it bound the Tfr most effectively. It is not possible to compare Pep-L and Pep-L* in these assays. Pep-L lacks a cysteine and therefore cannot be attached to the NPs. Pep-L* lacks a Fitc tag and cannot be tracked in FACS.

Line 301-302; Transcytosis of TfR antibodies is perhaps not only dependent on affinity. The avidity also seems to be an important issue as demonstrated by several papers from Ryan Watts group and Per-Ola Fresgaard group. Avidity might be a factor an important for nanoparticles and it will be a nice aspect to include in the discussion. The Bien-Ly reference (34) from Ryan Watts group made this point in the original version. Two additional references have been added and an additional paragraph on this subject is included in the discussion.

Not all the peptides could be inhibited by Tf. Have the authors any idea whether these peptides can bind to other receptors? All the active peptides were isolated by binding to the TfR. They have not been formally tested for their ability to bind other receptors

Pep10 has been published in a local thesis. Are there any evidence that this peptide in fact binds to TfR or is it just an assumption? If there are could some of these data be included in this paper? Pep-10 was isolated by its ability to bind both mouse and human Tfr in vitro, and tested for binding in an ELISA-type assay. Reference 22 – available on Open access.

Reviewer #3: 

1. For the uptake assays via endocytosis, the authors did not describe how the background noise was subtracted. A control peptide (such as a scrambled peptide) should be included in the experiments to determine the random uptake of the peptide by the endothelial cells in comparison with specific TfR-mediated uptake. The dextran control is used for non-specific endocytosis and the values in figure 1B are compared to this. Two peptides, R1 and R2 with similar size and charge to the active peptides (supplementary table 1), showed no significant difference from the dextran-endocytosis control. Although we agree that using scrambled peptides would be a better control, the results indicate that non-specific endocytosis is very limited and not significantly different from the (no reagent) negative control.

2. Binding of the polypeptides to TfR: the authors did not provide any direct evidence that their polypeptides can physically interact with and bind to TfR although transferrin was shown to inhibit endocytosis of some polypeptides tested. The authors may need to consider performing SPR analysis or cross-link of the polypeptides to TfR or Co-IP with TfR to determine whether their polypeptides indeed physically bind to TfR. See note above – all the active peptides, Pep1, Pep2 Pep10 and Pep-L were obtained by direct binding and selection of peptide libraries on pure human or mouse TfR in repeated rounds of selection followed by binding assays. Refs 21 and 22. We can therefore be certain that they bind TfR. 

3. Transcytosis assay: It is well known that the in vitro BBB models constituted by CMEC/D3 or B.end3 cell lines are very leaky. The authors did not describe how the collagen gel model was performed and whether there was any control molecule or control peptide to control the permeability and leakage of the model used. As noted above, more data on the hydrogel model has been included together with 3 references on its characteristics in nanoparticle transport studies .

4. A minor point is that the authors said in the manuscript that TfR is specifically expressed in brain endothelial cells. This is not true. A recent paper published in FBCNS analyzed the expression patterns of TfR and other RMT receptors in human and mouse brain vessels and peripheral tissues/vessels. The study found that TfR is also expressed in peripheral tissues and vessels. Thus, it is more accurate to say that TfR is more enriched in brain endothelial cells or expressed at a higher level in brain endothelial cells. This point was made in the Introduction and is supported by ref. 10.

---

## [Decision Letter · Decision Letter 1]

14 May 2021

Comparison of polypeptides that bind the transferrin receptor for targeting gold nanocarriers.

PONE-D-21-03180R1

Dear Dr. Male,

We’re pleased to inform you that your manuscript has been judged scientifically suitable for publication and will be formally accepted for publication once it meets all outstanding technical requirements.

Kind regards,

Mária A. Deli, M.D., Ph.D.

Academic Editor

PLOS ONE

Additional Editor Comments (optional):

Reviewers' comments:

Reviewer's Responses to Questions

**Comments to the Author**

1. If the authors have adequately addressed your comments raised in a previous round of review and you feel that this manuscript is now acceptable for publication, you may indicate that here to bypass the “Comments to the Author” section, enter your conflict of interest statement in the “Confidential to Editor” section, and submit your "Accept" recommendation.

Reviewer #1: All comments have been addressed

Reviewer #2: All comments have been addressed

Reviewer #3: All comments have been addressed

2. Is the manuscript technically sound, and do the data support the conclusions?

Reviewer #1: Yes

Reviewer #2: Yes

Reviewer #3: Yes

3. Has the statistical analysis been performed appropriately and rigorously? 

Reviewer #1: Yes

Reviewer #2: Yes

Reviewer #3: Yes

4. Have the authors made all data underlying the findings in their manuscript fully available?

Reviewer #1: Yes

Reviewer #2: Yes

Reviewer #3: Yes

5. Is the manuscript presented in an intelligible fashion and written in standard English?

Reviewer #1: Yes

Reviewer #2: Yes

Reviewer #3: Yes

6. Review Comments to the Author

Reviewer #1: The authors have adequately addressed my comments raised in a previous round of review and I feel that this manuscript is now acceptable for publication.

Reviewer #2: The manuscript would have benefitted from more kinetic experiments and perhaps some visual imaging demonstration uptake. But the authors have addressed the comments and I have no further comments.

Reviewer #3: Thanks for the revisions. The questions have been addressed. I am fine with the revised manuscript.

7. PLOS authors have the option to publish the peer review history of their article (what does this mean?). If published, this will include your full peer review and any attached files.

Reviewer #1: No

Reviewer #2: No

Reviewer #3: **Yes: **Wandong Zhang

---

## [Editor Report · Acceptance letter]

18 May 2021

PONE-D-21-03180R1 

Comparison of polypeptides that bind the transferrin receptor for targeting gold nanocarriers. 

Dear Dr. Male:

I'm pleased to inform you that your manuscript has been deemed suitable for publication in PLOS ONE. Congratulations! Your manuscript is now with our production department. 

Kind regards, 

on behalf of

Dr. Mária A. Deli 

Academic Editor

PLOS ONE